# A Study on an Organic Semiconductor-Based Indirect X-ray Detector with Cd-Free QDs for Sensitivity Improvement

**DOI:** 10.3390/s20226562

**Published:** 2020-11-17

**Authors:** Jehoon Lee, Hailiang Liu, Jungwon Kang

**Affiliations:** Department of Electronic and Electrical Engineering, Dankook University, Gyeonggi-do 16890, Korea; usyj0512@gmail.com (J.L.); liuhailiang107@gmail.com (H.L.)

**Keywords:** indium phosphide quantum dot, photodetector, indirect X-ray detection

## Abstract

In this paper, we studied the optimized conditions for adding inorganic quantum dots (QD) to the P3HT:PC_70_BM organic active layer to increase the sensitivity of the indirect X-ray detector. Commonly used QDs are composed of hazardous substances with environmental problems, so indium phosphide (InP) QDs were selected as the electron acceptor in this experiment. Among the three different sizes of InP QDs (4, 8, and 12 nm in diameter), the detector with 4 nm InP QDs showed the highest sensitivity, of 2.01 mA/Gy·cm^2^. To further improve the sensitivity, the QDs were fixed to 4 nm in diameter and then the amount of QDs added to the organic active layer was changed from 0 to 5 mg. The highest sensitivity, of 2.26 mA/Gy·cm^2^, was obtained from the detector with a P3HT:PC_70_BM:InP QDs (1 mg) active layer. In addition, the highest mobility, of 1.69 × 10^−5^ cm^2^/V·s, was obtained from the same detector. Compared to the detector with the pristine P3HT:PC_70_BM active layer, the detector with a P3HT:PC_70_BM:InP QDs (1 mg) active layer had sensitivity that was 61.87% higher. The cut-off frequency of the P3HT:PC_70_BM detector was 21.54 kHz, and that of the P3HT:PC_70_BM:InP QDs (1 mg) detector was 26.33 kHz, which was improved by 22.24%.

## 1. Introduction

Colloidal semiconductor nanocrystals (NCs), such as quantum dots (QDs), nanowires (NWs), and nanoplatelets (NPLs), are being widely studied [1,2,3,4]. The NCs have the advantage that they can adjust optical and electrical characteristics by changing their properties, such as size, shape, and composition of NCs, and thus have been applied to various fields, such as light-emitting diodes (LEDs) [5,6], photodetectors [7,8], biomedical imaging [9,10], and radiation [11,12]. Among them, in the field of radiation detectors, there is a growing trend of research on mixing them with organic semiconductors and NCs, such as hybrid or additives architecture, to improve detection sensitivity [13,14]. In the organic/NC blending, the organic semiconductors have the advantage of being easy to process, low in cost, and applicable to flexible devices, because the solution process is applied to the manufacturing. The NCs are expected to improve device performance by being mixed with organic semiconductors because of their excellent chemical and physical stability, excellent carrier mobility, and efficient photon absorption on the organic polymer interface. The NCs also offer adjustable bandgaps and high quantum efficiency benefits and overcome the charge–transport pathway limits of organic semiconductor interfaces by forming additional percolation paths. The various types of NCs, such as QDs, NWs, and NPLs, are used for radiation detectors and consist of common semiconductor materials, such as group II–VI (e.g., cadmium selenide (CdSe) [15,16,17] and cadmium telluride (CdTe) [18,19]), IV–VI (e.g., lead sulfide (PbS) [20]), and III–V (e.g., indium phosphide (InP) [21] and gallium arsenide (GaAs) [22]). Metal oxide NCs have been applied to direct-type radiation detectors. Among the metal oxide NCs, Bi_2_O_3_ with a high atomic number has been reported to have the stopping power for high energy photons by being applied to the conversion layer [23,24]. Commonly used NCs are composed of hazardous substances with environmental problems, such as cadmium (Cd) and lead (Pb) [25,26]. Therefore, research on Cd- and Pb-free NCs has become more important, and use of In-based NCs, such as copper indium sulfide (CuInS_2_) [27], indium phosphide (InP) [28], and indium zinc phosphide (InZnP) [29] tends to increase in photodetector, because they are quite similar to the Cd-based NCs in terms of optical properties and colloidal stability [30]. In addition to the photodetectors, since the X-ray detector is similar to the photodetection method, Cd- and Pb-free NCs could be applied to the X-ray detector, which is in close contact with the human body; so the use of Cd- and Pb-free NCs is further required.

Two types of detection methods have been developed: one is indirect detection [31,32] and the other is direct detection [33,34]. In direct detection, the absorbed X-ray photons are directly converted to electron–hole pairs in the photoconductive conversion layer, and then generated charges are collected at each biased electrode. On the other hand, the indirect detection is combined with a scintillator, which absorbs X-ray photons, converts them into visible photons, and then generates electron–hole pairs because of the incident visible photons in the active layer of the photodetector. In this study, we investigated the organic semiconductor-based indirect detector with InP QDs to improve the performance of the X-ray detector. The active layer was composed of conjugated polymer poly(3-hexylthiophene) (P3HT) as the electron donor, and fullerene derivatives [6,6]-phenyl-C71-butyric acid methyl ester (PC_70_BM) as the electron acceptor. In addition, InP QD was introduced, because it delivers additional energy to the P3HT donors through the Förster resonant energy transfer (FRET) effect to create additional carriers [35,36]. The InP QDs also had a light emission, well matched with the P3HT:PC_70_BM active layer absorbance wavelength. The generated emission transfers photon energy to the active layer and creates additional charge carriers, which improve the performance of the detector. The absorbance of the P3HT:PC_70_BM active layer was also improved by the addition of InP QDs. In addition to improving energy transfer and absorbance, adding InP QDs could facilitate carrier transport because of the small difference of the highest occupied molecular orbital (HOMO) energy levels between the P3HT:PC_70_BM organic material and InP QDs [37].

Figure 1a depicts the structure of the detector and the mixed state of the active layer, and Figure 1b shows the corresponding energy-band diagram of the materials that compose the detector, which consisted of two opposing electrodes, defined as an indium–tin–oxide (ITO)-anode and a LiF/Al -cathode. For the hole–transport layer (HTL), poly(3,4-ethylenedioxythiophene):poly (styrene sulfonate) (PEDOT:PSS) was used and spin-coated onto the ITO anode. The active layer was composed of P3HT, PC_70_BM, and InP QDs (P3HT:PC_70_BM:InP QDs) and formed a bulk heterojunction (BHJ) on the PEDOT:PSS layer. The visible light converted by means of the scintillator was irradiated in the BHJ active layer, and excitons were separated into electrons and holes at the boundary between the donor and the acceptor. In the fabricated X-ray detector, it could be applied as a photodetector if the scintillator was separated. Therefore, by evaluating the characteristics, such as series resistance (R_s_) and short-circuit current (J_sc_) parameters of the photodetector using an artificial solar radiation simulator, the characteristics of the radiation detector combined with a scintillator could be evaluated in advance. The R_s_ and J_sc_ were extracted from the *J–V* curve of the photodetector, which can be used to verify that it works as a detector. Here, J_sc_ is the current density measured by incident light when the detector was shorted and could be extracted from the 0 V bias of the *J–V* curve. The R_s_ represents a resistance across the substrate, active layer, and electrode and was measured in reverse slope when the detector was on open circuit with no current flowing in the *J–V* curve. In addition, X-ray detector characteristics were extracted from the third quadrant region, which is reverse-biased [38]. Therefore, the X-ray detector performances could be evaluated in advance by means of the J_sc_. After the photodetector evaluation, it was combined with a scintillator to evaluate the detector characteristics, such as collected current density (CCD), dark current density (DCD), and detection sensitivity under X-radiation. In addition to detection sensitivity, frequency response is also an important characteristic in X-ray imaging applications, because high-resolution X-ray images can be extracted at low dose rates by means of the high sensitivity and high frequency response. Green light-emitting diode (LED) pulsed illumination was used to evaluate the frequency response of the detector with the scintillator separated.

## 2. Experimental Detail

### 2.1. Detector Fabrication

Figure 2 shows the fabrication sequence of the proposed detector with a P3HT:PC_70_BM:InP QDs active layer. After indium–tin–oxide (ITO) was patterned on glass substrate applied as the anode electrode, a 2 × 2 mm^2^ active area of each cell was formed with insulators (e.g., polyimide). The hole–transport layer (HTL) was spin-coated on the ITO at 3000 rpm with a conductive organic material such as poly(3,4-ethylenedioxythiophene):poly (styrene sulfonate) (PEDOT:PSS) and baked at 150 °C for 30 min. The thickness of the HTL was about 30 nm. To prepare the P3HT:PC_70_BM:InP QDs solution, first, the P3HT and PC_70_BM were mixed by a weight ratio of 1:1, and a total weight of 20 mg was dissolved in 1 mL of chlorobenzene. Then, the desired size and amount of InP QDs were precipitated through a centrifuge and dissolved in the prepared P3HT:PC_70_BM solution. First, the experiment was conducted according to the three sizes of 4, 8, and 12 nm diameter QDs in the active layer, the amount was fixed at 3 mg to check the size effect. In addition, in order to confirm the optimal active-layer solution state, after the size was fixed, the additional amounts of QDs from 0 to 5 mg in active-layer solution were added. The InP QDs added in P3HT:PC_70_BM active-layer solutions were spin-coated on HTL at 700 rpm for 30 s, and then baked 150 °C for 10 min. The cathode electrode consisted of lithium fluoride (LiF) and aluminum (Al) was deposited to a thickness of 0.5 and 120 nm, respectively, by thermal evaporation under a pressure of 10^−7^ torr. Finally, the detector was encapsulated with a glass cover to keep out oxygen and humidity.

### 2.2. Experimental Setup

The experimental setup for evaluating the characteristics of the X-ray detector is shown in Figure 3. The setup consists of an X-ray generator (AJEX 2000H), a solar illumination simulator (San Ei Elec. XES-40S2-CE), and an electrometer (Keithley 2400) for measuring the current depending on the applied bias and exposed conditions. The fabricated detector was evaluated in two ways. One was with the detector separated by a CsI(Tl) scintillator, and the other was the detector combined with a CsI(Tl) scintillator. The detector without the CsI(Tl) scintillator was exposed to light from an AM 1.5 G filtered Xe lamp in a solar-illumination simulator. The distance between the solar simulator and the detector was 25 cm for 100 mW/cm^2^ light exposure. The generated charge was collected by applying a bias of −1.0 to 1.0 V. The parameters obtained during the artificial solar irradiation were short-circuit current density (J_sc_) and series resistance (R_s_) calculated from the *J–V* characteristics.

The performance of the X-ray detector coupled with the CsI (Tl) scintillator (Hamamatsu Photonics J13113), consisting of 0.5 mm of Al and 0.4 mm of CsI (Tl), was evaluated under X-radiation. For all experiments, the operating condition of the X-ray generator was fixed at 80 kV_p_ and 60 mA·s and radiation for 1.57 s. The distance between the X-ray generator and the detector was fixed at 30 cm. The exposed X-ray dose was measured using an ion chamber (Capintec CII50) at the same distance as the detector. The absorbed dose was converted from the exposure dose, which was measured using the ion chamber. Under the same conditions, the absorbed dose was 3.44 mGy. A bias of −0.2 to −1.4 V was applied between the electrodes to collect the current generated during X-ray exposure. The collected current density (CCD) during X-ray on state and dark current density (DCD) during X-ray off state were calculated by Equations (1) and (2), respectively. Sensitivity was calculated using Equation (3), which represents the relationship of generated current divided by absorbed dose.
(1)CCD [μAcm2]= Collected Current during X−ray ONExposed Detection Area
(2)DCD [μAcm2]= Collected Current during X−ray OFFExposed Detection Area
(3)Sensitivity [μAmGy·cm2]= CCD−DCDAbsorbed Dose

## 3. Result and Discussion

### 3.1. Experiment on QD Size Change

First, we investigated the change in detector characteristics with the change in InP QD size. The absorbance spectra of P3HT:PC_70_BM thin film and P3HT:PC_70_BM thin films with various sizes of InP QDs (4, 8, and 12 nm diameters) were measured using UV/Vis spectrometer (OPTIZEN 2120UV), as shown in Figure 4a. The absorbance of the P3HT:PC_70_BM blended with different sizes of InP QDs was higher than the absorbance of the pristine P3HT:PC_70_BM film, because QDs have a high extinction coefficient (k ≈ 0.46 @ 510 nm). Since the absorbance characteristics were affected by the QD distribution, the distance between the QD particles in the P3HT:PC_70_BM active layer was calculated assuming that the same weight of InP QD was well-dispersed in the P3HT: PC_70_BM active layer [39]. The added InP QD weight was fixed at 3 mg and the volume percentage in the P3HT:PC_70_BM active layer was 4.05% (see Appendix A for details on calculating the QD percentage and distance between QDs). As the InP QD sizes changed to 4, 8, and 12 nm, the distances between QDs in the same volume were 5.56, 11.11, and 16.65 nm, respectively. Therefore, the shortly distributed 4 nm InP QDs showed the highest absorbance in the P3HT:PC_70_BM active layer. In addition, the emission spectrum of the CsI(Tl) scintillator showed the maximum peak at 550 nm, which was well matched with the absorbance of the P3HT:PC_70_BM:InP QDs film. Before evaluating the X-ray detector performance, the *J–V* characteristics of the detector without the CsI(Tl) scintillator were obtained using the solar simulator, as shown in Figure 4b. The extracted parameters such as J_sc_ and R_s_ were calculated from the *J–V* curve of the detectors contained different sizes of QDs. As shown in the table in Figure 4b, the R_s_ of the P3HT:PC_70_BM:InP QDs (12 nm) detector was higher than the pristine P3HT:PC_70_BM detector. As the size of the added QD increased, defects in the active layer could increase or contact between layers could become poor. Since the defect density could affect the R_s_, the defect density of the fabricated detectors was calculated (refer to Appendix A for details on calculating the defect density). Depending on the size of the added QDs (4, 8, and 12 nm), the defect densities were calculated as 6.23, 6.07, 6.18, and 6.38 × 10^15^ cm^−3^, respectively. The detector with the P3HT:PC_70_BM:InP QDs (12 nm) active layer showed higher defect density than the pristine detector with the P3HT:PC_70_BM active layer. The detector with 4 nm InP QDs blended in the P3HT:PC_70_BM layer showed the highest J_sc_ of 9.4 mA/cm^2^ and the lowest R_s_ of 362.01 Ω. The addition of InP QDs in the P3HT:PC_70_BM active layer tends to increase the J_sc_, because the absorbance increases, resulting in improved charge generation, and R_s_ increases as the InP QD size changes, because the morphology and percolation path of the P3HT:PC_70_BM:InP QDs active layer deteriorates.

Next, in order to measure the charge generated by X-ray radiation, an experiment was conducted by combining the detector and the scintillator. As shown in Figure 3, the operating conditions of X-ray generator were fixed for all experiments. The detectors were fabricated under four conditions with the pristine P3HT:PC_70_BM and P3HT:PC_70_BM:InP QDs (4, 8, and 12 nm diameter) active layers. The voltage applied to X-ray generator operation was applied in the range of −0.2 to −1.4 V. The P3HT:PC_70_BM detector sensitivity saturation point was −0.6 V (blue dash box), and the P3HT:PC_70_BM:InP QDs detector tended to saturate at −1.0 V (red dash box), as shown in Figure 5a. The experiment on the linearity of the detector is shown in Figure 5b, and the applied voltage of the detector was determined by means of the experiment in Figure 5a. An experiment related to the change in CCD–DCD according to the change in absorbed dose (0.55 ~ 5.46 mGy) was conducted using the detector with P3HT:PC_70_BM:InP QDs (4 nm diameter) active layer, which showed the best characteristics. As shown in Equation (3), the two values of CCD–DCD and absorbed dose must have a linear relationship, the coefficient of determination (R-square) of a linear relationship was 0.96, and the slope of the fitted line was 2.04 mA/Gy·cm^2^, which was similar to the measured sensitivity. Figure 5c shows the X-ray parameters, such as CCD, DCD, and sensitivity according to the InP QD sizes, and applied voltage condition was fixed at −1.0 V. As shown in Figure 4a, the overlapped region of the P3HT:PC_70_BM:InP QDs (4 and 8 nm) absorbance curves and the scintillator emission curve related to carrier generation showed similar areas. Therefore, the sensitivities in both cases showed similar values. As shown in Figure 4b, the R_s_ value of the fabricated detector was the lowest at 4 nm InP QDs and 423.93 Ω with 8 nm InP QDs. Since the R_s_ was related to the carrier transport, the detector with 4 nm InP QDs showed slightly higher sensitivity than the detector with 8 nm InP QDs. According to that, when InP QDs were added in P3HT:PC_70_BM active layer, DCD slightly increases, but CCD tends to increase significantly by at least 24.24%. Therefore, the detection sensitivity (proportional to CCD–DCD) was increased, and the highest detection sensitivity, of 2.01 mA/Gy·cm^2^, was shown at the P3HT:PC_70_BM:InP QDs (4 nm diameter) detector and was improved by 44.60%.

### 3.2. Experiment on QD Amount Change

In order to improve the X-ray detector performance, experiments were conducted with InP QD in the content range of 0 to 5 mg in the P3HT:PC_70_BM active layer. The InP QD of the active layer was fixed to 4 nm in diameter, which showed excellent properties in the previous experiment. The atomic force measurement (AFM) image in Figure 6a shows that the surface roughness of the pristine P3HT:PC_70_BM and P3HT:PC_70_BM:InP QDs (1, 3, and 5 mg) films. The average surface roughness (R_q_) increased with increasing InP QDs content in the active layer. As R_q_ increased, the surface area that effectively absorbed light increased, but as R_q_ continued to increase as QD was added, the contact between the active layer and the electrode became poor. As the amount of QDs increased, the aggregation site interfered with charge transfer and the network between InP QDs and P3HT:PC_70_BM became unstable. In addition, the added QDs affected the crystallinity of the conjugated organic semiconductor and changed the charge transport characteristics [40]. Figure 6b shows the R_s_ and X-ray parameters, such as CCD and sensitivity. With the addition of InP QDs, the R_s_ of the fabricated detector decreased and the CCD and sensitivity increased. The CCD and sensitivity with a tendency opposite to that of the R_s_ exhibited a higher value than pristine P3HT:PC_70_BM, because the surface area absorbing light increases and the generated charge transfer were facilitated by the low R_s_. Therefore, the detector with the P3HT:PC_70_BM:InP QDs (1 mg) showed the highest CCD, of 8.96 μA/cm^2^, and the highest sensitivity, of 2.25 mA/Gy·cm^2^, showing that the sensitivity was improved by 61.87% over that the P3HT:PC_70_BM detector. The lowest R_S_ of 268.05 Ω appeared, because 1 mg of InP QDs was evenly dispersed and helped the charge transfer in the active layer.

The charge mobility of organic conjugated polymers shows anisotropic characteristics; so it is mainly used in organic light-emitting diode (OLED) and organic photovoltaics (OPV) devices that transfer charges vertically to the cathode and anode. In general, conjugated polymers have higher charge mobility in the π orbital direction than in other directions [41]. In sandwich-type devices including OPV, the charge mobility in the vertical direction to the electrodes has a desired correlation with the performance of the device; so the mobility of the vertical charge should be measured by means of a method such as time-of-flight (TOF), charge extraction by linearly increasing voltage (CELVIC), or space-charge-limited current (SCLC). Among them, mobility is measured by the CELIV or SCLC methods, which are widely used for analyzing the charge mobility of the bulk-heterojunction solar-cell (BHJSC) active layer, since the TOF measurement needs the thickness of the active layer to be at least 1 μm. However, SCLC measurement is the simplest way to measure the *J–V* curve in the dark, since it can directly calculate the basic transport parameters better than the CELIV [42]. Therefore, the carrier mobility of the P3HT:PC_70_BM and P3HT:PC_70_BM:InP QD (1 to 5 mg) detector was obtained using the SCLC model from the *J–V* characteristics in the dark in Figure 7a [43]. Theoretical calculations and experimental result indicate [44] that the charge injection barrier should not exceed 0.3 eV in SCLC model and should be coated with a PEDOT:PSS thin film of about 50 nm [45,46], which was similar to the conditions of the detector being used in the experiment. According to the Mott–Gurney law, the current density could be expressed as Equation (4) in Trap-limited SCLC region [47], because the organic-based detector is experimentally difficult to reach the trap-free limit due to the trap states being widely distributed in the energy band [48,49]. The SCLC region is defined as the region with a slope 2 in the graph, where the X and Y axes are log scale, shown in Figure 7b.
(4)$$J=\frac{9}{8}\cdot \mu \cdot \theta \cdot \frac{V_{a}^{2}\cdot \varepsilon_{0}\cdot \varepsilon_{r}}{L^{3}}$$
where *μ* is the mobility, θ is the ratio of free carriers among total carriers, ε_0_ is the free-space dielectric constant (8.85 × 10^−12^ F/m), and ε_*r*_ is the relative dielectric constant of the active layer. The relative dielectric constant of P3HT:PC_70_BM:InP QDs mixed layer was calculated using the dielectric constant equation of mixed material [50]. *V_a_* is the voltage drop across the detector, *J* is the current density, and *L* is the thickness of the active layer, measured using a surface profiler (KLA-Tencor AS-500). Due to the small amount of QDs added, all active layers were about 150 nm thick. Figure 7b is the log scale J–V graph to extract the mobility in terms of the condition of P3HT:PC_70_BM:InP QDs (1 mg) active layer. All the calculated mobilities as a function of the InP QDs content in the active layer are listed in the table in Figure 7b. The mobility was calculated sequentially as 0.519, 1.69, 1.19, and 0.958 × 10^−5^ cm^2^/V·s depending on the content of InP QDs (0, 1, 3, and 5 mg) in the active layer. The added QDs formed a charge–transport path, improving the mobility compared to that of the pristine P3HT:PC_70_BM detector. The detector with the P3HT:PC_70_BM:InP QDs (1 mg) active layer showed the highest mobility, of 1.69 × 10^−5^ cm^2^/V·s, because the charge transfer and collection of generated charges increased in relation to the result of increasing the surface area of light absorption and the lowest R_S_.

### 3.3. Changes in Frequency Response According to QD Addition

Along with the previously obtained sensitivity, the frequency response is an important parameter in the imaging application of the X-ray detector. For the frequency response experiment, the optimized condition of the P3HT:PC_70_BM:InP QDs (1 mg) active layer obtained from previous experiments were applied to the X-ray detector. The apparatus for the frequency response experiment is depicted in Figure 8a. A green LED array with a 540 nm peak similar to the CsI (T1) scintillator emission peak used in the previous experiment was used, and the intensity of the LED array was fixed at 60 μw/cm^2^. A function generator (RIGOL DG1022Z) was used to generate green pulsed light in the range of 2 to 100 kHz (@ 50% duty cycle) in the LED array, which was exposed to the P3HT:PC_70_BM:InP QDs (1 mg) detector. The signal amplitude at the lowest frequency of 2 Hz (U_0_) and the signal amplitude at 1 kHz (U) are shown in Figure 8b. The signal was acquired using an oscilloscope (LeCroy 104Xi). The ratio of U_0_ and U was extracted while changing the frequency from 2 to 100 kHz, and the amplitude bode plot was plotted by taking 20 log of the ratio of U_0_ to U as shown in Figure 8c. The cut-off frequency corresponds to the −3 dB line in the amplitude bode plot. The detector with the P3HT:PC_70_BM and P3HT:PC_70_BM:InP QDs (1 mg) active layer showed a cut-off frequency of 21.54 and 26.33 kHz, respectively. When the test was done with the same conditions, a silicon photodetector (Hamamatsu S3590-08) measured a cut-off frequency of 53 kHz. The detector with P3HT:PC_70_BM:InP QD (1 mg) active layer had about 50% of the cut-off frequency compared with that of the silicon photodetector, but since the addition of InP QDs improved the optical and electrical properties of the detector, it shows a higher cut-off frequency that is 22.24% higher than the pristine P3HT:PC_70_BM detector.

## 4. Conclusions

In this paper, we studied the method of adding inorganic QDs to the organic active layer in order to improve the detection sensitivity of the indirect X-ray detector. Adding QDs was expected to improve detector performance because of their excellent chemical and physical stability, excellent carrier mobility, and efficient photon absorption on the organic polymer interface. The organic active layer consisted of a conjugated polymer P3HT and fullerene derivatives PC_70_BM. Commonly used QDs are composed of hazardous substances with environmental problems, so InP QDs were selected as the electron acceptor in this experiment. To improve the sensitivity of the detector, experiments were performed to find the optimal conditions by changing the size and amount of InP QDs added into the P3HT:PC_70_BM active layer. After the amount of additive QDs were fixed to 3 mg, the series resistance (R_s_), collected current density (CCD), and sensitivity of the detector were extracted during X-ray exposure according to the QD size change (4 to 12 nm in diameter). As the QDs size increased, the R_S_ increased, the CCD decreased, and the sensitivity decreased. If the added QDs was larger than 4 nm, the absorbance of the active layer decreased, because the QDs were not well-dispersed in the organic active layer. Therefore, the detector with the P3HT:PC_70_BM:InP QDs (4 nm diameter) showed the highest sensitivity, of 2.01 mA/Gy·cm^2^, showing that the sensitivity was improved by 44.60% over that of the P3HT:PC_70_BM detector. To further improve the sensitivity, the QDs size was fixed to 4 nm in diameter, and then the amount of QDs added to the organic active layer was changed from 0 to 5 mg. The highest sensitivity, of 2.26 mA/Gy·cm^2^, was obtained from the detector with P3HT:PC_70_BM:InP QDs (1 mg) active layer. In addition, the highest mobility, of 1.69 × 10^−5^ cm^2^/V·s, was obtained from the same detector. Compared with the pristine detector with P3HT:PC_70_BM active layer, the detector with P3HT:PC_70_BM:InP QDs (1 mg) active layer had the sensitivity that was 61.87% higher. The changed amount of InP QDs in the P3HT:PC_70_BM organic active layer changed the surface morphology, thereby increasing the charge generations by improving the light-absorbing surface area. In addition, the well-formed network between inorganic InP QDs and organic P3HT:PC_70_BM active layer were reduced R_S_ and improved the mobility, because it helps charge transfer and charge collection. The frequency response of the detector without the scintillator was evaluated using a pulsed green LED (emission peak at 540 nm). The cut-off frequency, extracted from the amplitude bode plot, of the P3HT:PC_70_BM detector was 21.54 kHz and that of the P3HT:PC_70_BM:InP QDs (1 mg) detector was 26.33 kHz. Although the detector with P3HT:PC_70_BM:InP QDs (1 mg) active layer cut-off frequency was about 50% compared with the silicon photodetector, it was confirmed that the cut-off frequency was improved by 22.24% over that of the pristine P3HT:PC_70_BM detector.

## Figures and Tables

**Figure 1 sensors-20-06562-f001:**
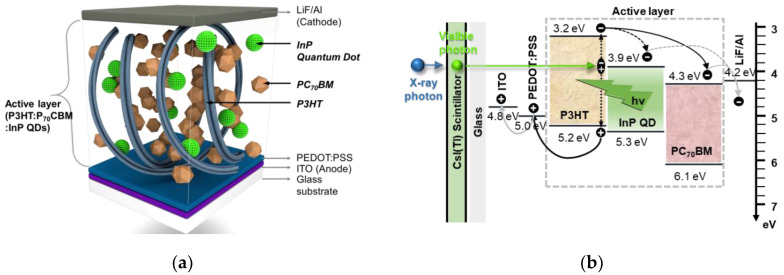
(**a**) The heterojunction structure of detector with P3HT:PC_70_BM:InP quantum dots (QDs) image and (**b**) schematics diagram with corresponding energy band diagram.

**Figure 2 sensors-20-06562-f002:**
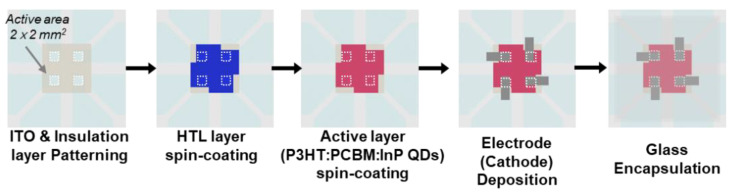
Fabrication sequence of InP QDs contented with P3HT:PC_70_BM active layer for radiation detector.

**Figure 3 sensors-20-06562-f003:**
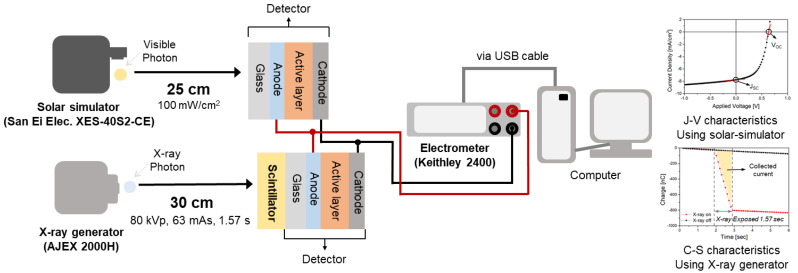
Schematics diagram of experimental setup for measuring the solar parameters (J_SC_, R_S_, and power conversion efficiency (PCE)) and X-ray parameters (collected current density (CCD), dark current density (DCD), and sensitivity).

**Figure 4 sensors-20-06562-f004:**
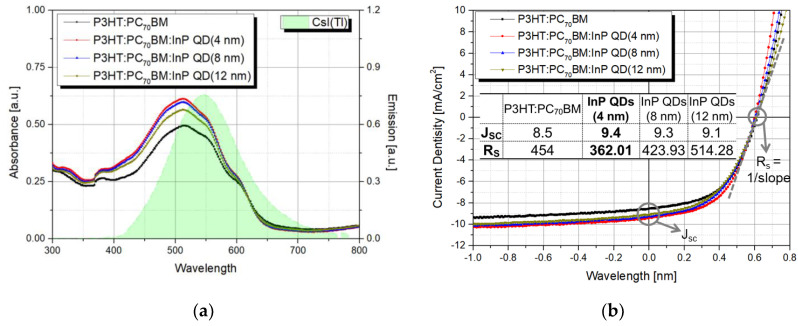
(**a**) Absorbance and emission spectra of CsI(Tl) scintillator and (**b**) *J–V* curve according to InP QDs size (4, 8 and 12 nm) with P3HT:PC_70_BM active layer.

**Figure 5 sensors-20-06562-f005:**
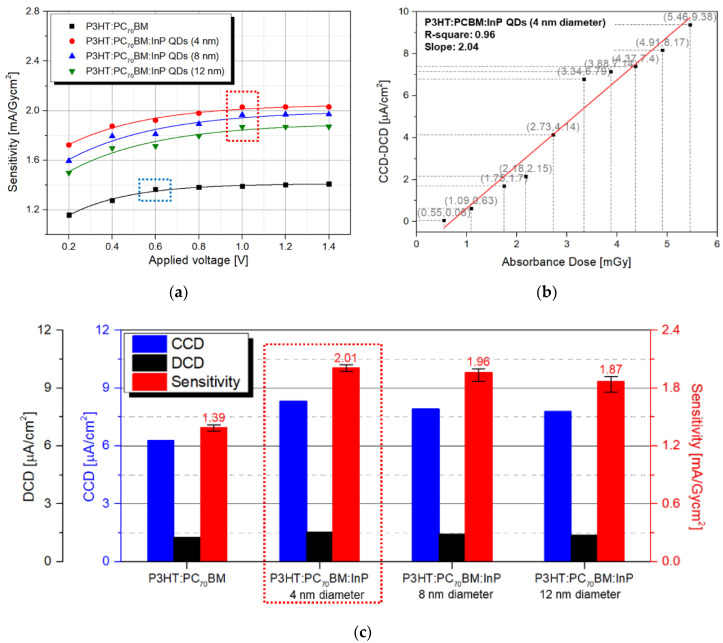
(**a**) A graph of the change in detector sensitivity according to applied voltage and (**b**) CCD–DCD value of the detector with P3HT:PC_70_BM:InP QDs (4 nm diameters) depending on the absorbed dose and (**c**) X-ray parameters, such as DCD, CCD, and sensitivity according to the pristine P3HT:PC_70_BM and P3HT:PC_70_BM:InP QDs (4, 8, and 12 nm diameter).

**Figure 6 sensors-20-06562-f006:**
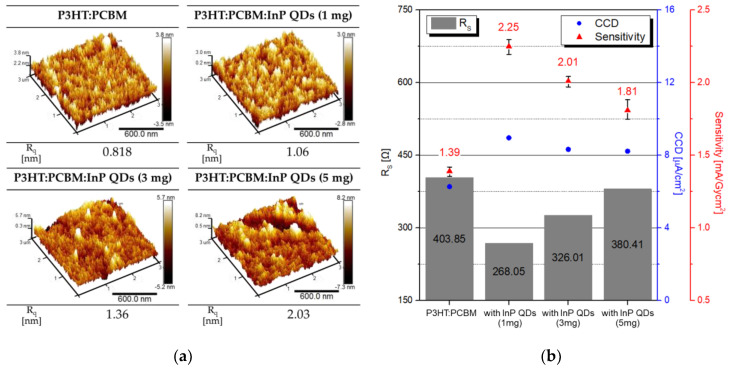
(**a**) The atomic force measurement (AFM) image and R_q_ depending on the P3HT:PC_70_BM:InP QDs (0, 1, 3, and 5 mg) films and (**b**) the parameters of R_S_, CCD, and sensitivity of detector for different InP QD amounts.

**Figure 7 sensors-20-06562-f007:**
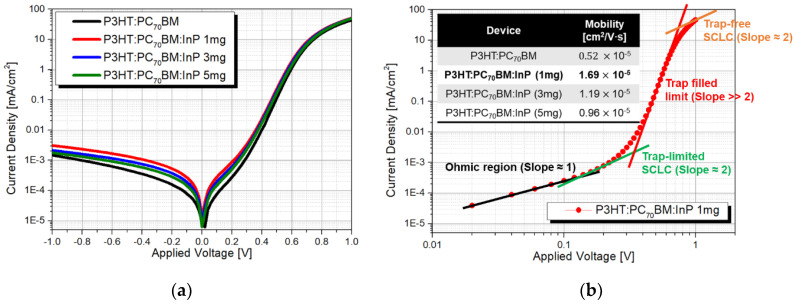
(**a**) *J–V* characteristics of detector with P3HT:PC_70_BM and P3HT:PC_70_BM:InP QDs (1, 3, 5 mg amount) under dark conditions and (**b**) the space charge limited current (SCLC) behavior of the detector with P3HT:PC_70_BM:InP QDs (1 mg).

**Figure 8 sensors-20-06562-f008:**
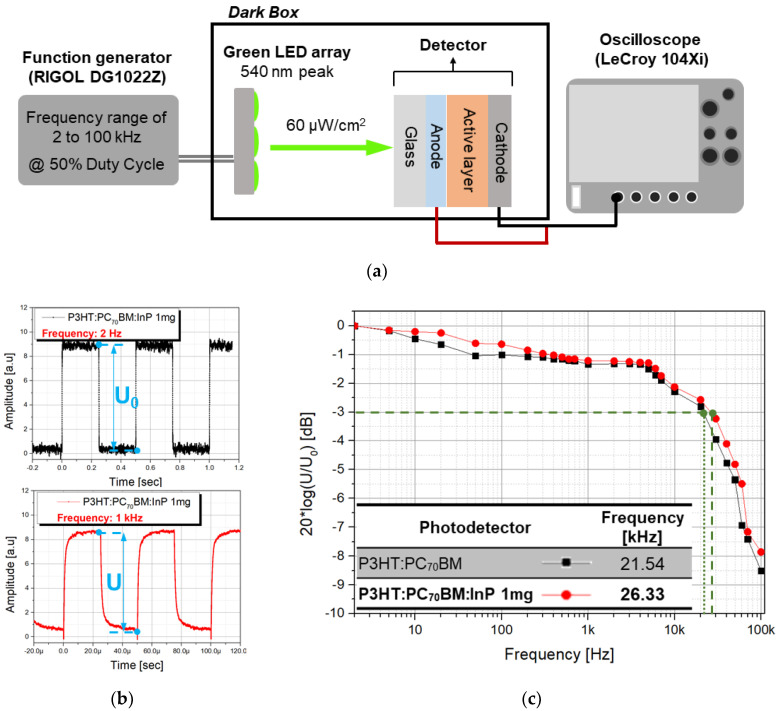
(**a**) Frequency response experiment setup, (**b**) square light pulse at 2 and 1 kHz, and (**c**) extracted amplitude bode plot.

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
