# Peer review of "A Study on an Organic Semiconductor-Based Indirect X-ray Detector with Cd-Free QDs for Sensitivity Improvement"

_sensors, 2020, doi:10.3390/s20226562_

Round 1
Reviewer 1 Report
In this manuscript the authors study P3HT:PCBM blends with embedded InP QDs to increase the sensitivity of the indirect X-ray detectors. The authors made a screening upon the concentration and size of the QDs and found the best conditions to increase the sensitivity of the detector up to 61.87%. Moreover the frequency response of the detector without scintillator was evaluated resulting in a cut-off frequency improved by 22% compared the system without QDs. The study is of interest for the scientific community, however major revisions could improve the current form of the manuscript.
- Introduction: “NCs such as Bi2O3 with a high atomic number [23, 24] are used in order to have the stopping power for high radiation energy.” Please improve this sentence to explain why the authors want to underline this peculiarity of the Bi2O3 QDs.
- The InP could improve the optical properties of the P3HT:PCBM blend, however the author reported in Figure 4a only the absorbance in which the intensity increases upon the addition of the QDs because of its higher extinction coefficient (please report the value). Could the author report the steady state and time resolved photoluminescence to study also the carrier dynamics?
- The solvent of the solution is not specified in the experimental section, please add more details about the solution preparation and exactly how the authors add the QDs in the organic blend.
- The 4 nm size QDs work better because of the well-distribution. This is not demonstrated; a morphological and structural characterization is needed (SEM, HR-TEM and AFM) to have a comprehensive analysis of the QDs distribution. If not this sentence is not justified. Otherwise the authors should estimate the volume distribution, as in this ref ACS Energy Lett.2020, 5, 2, 418–427, for different size and different concentration of the QDs.
- In Figure 5c the sensitivity for 4 nm and 8nm are very similar. A standard deviation is indicated to be 1.96 and 2.01 respectively, is this difference significant? Please discuss in depth this point.
- The authors hypothesized that the aggregation interferes with charge transfer (page 6 line 202-205). This is a valid option, however the P3HT:PCBM blend could go through a different crystallization due to the presence of the QDs, which can change the transport properties. Please add also this hypothesis (Nature Mater12, 628–633 (2013)). Moreover the aggregation of the QDs is not experimentally demonstrated, so this sentence needs a revision.
- Please add how the thickness of the layer of 150 nm was measured.
- English errors need to be corrected.
Author Response
Dear Reviewer of Sensors
We responded point-by-point according to the reviewer's comments. The response file was attached, please see the attachment.
Best Regards
Prof. Jungwon Kang

Reviewer 2 Report
The manuscript reports on an improvement of the sensitivity and other detector properties of Organic Semiconductor-Based Indirect X-ray Detector by adding InP quantum dots into the active layer. The research is well substantiated and results are convincing and properly documented. The paper should be published. Nevertheless, reading the manuscript,
I noticed several items, which should be clarified, improved or better discussed before the manuscript may be accepted in Sensors. The respective list follows.
1. Table in Fig. 4(b): Authors properly comment their main achievement recommending InP QDs (4nm) at the optimum for the detector fabrication. Nevertheless, I noticed some inconsistency at the R_S of sample with InP QDs (12nm), which even exceeds the R_S of pure P3HT:PC70BM. Authors should comment this anomaly.
2. Fig. 5(a): Curves fitting experimental points resembles the Hecht equation fit routinely used at the characterization of semiconductor detectors for direct detection. Authors should define the curve used for the fit and optionally comment its shape with respect to the Hecht equation. Perhaps, some additional data about their sensor could be collected by this way.
3. In spite of well documented advantage of the sample with QD 4nm, I am not definitely convinced by presented results that the sample with larger QD 12nm would be worse, when a larger density of QDs 12nm was used. As one may deduce from Fig. 4(a), the lower confinement energy of QD 12 nm shifts the absorption to the lower wavelength, where the scintillator peak has the maximum. Consequently, if the QD 12 nm density was increased, the sensitivity could exceed the sample with QD 4 nm. The comparison of detectors with the same mass of InP QDs prioritizes the smaller QDs, which fill the detector at much larger density than the large ones. Authors should consider that the diameter appears in the third power at the QD density. The results could be different, if samples with the same QDs density were analyzed. Authors should comment this objection.
4. Authors argue that SCLC may be conveniently used for the determination of mobility through eq. (4) where an effect of traps may be neglected. I do not see a reason for such simplification. It is known that traps significantly affect SCLC in semiconductors. Evaluated mobility could be then burdened by an appreciable error.
5. Typing error: The word 'Expermenet' in the title of Section 3.2. is unknown to me. In addition, several faults in grammar were detected, for example missing verb in line 223. There are several very long sentences in Section 3.2. I recommend to split them to shorter statements.
Author Response

(The authors gave the same response as above.)

Round 2
Reviewer 1 Report
The revisions improved the manuscript and it should be accepted as it is. Some typing errors can be corrected during the proof revisions.
Author Response
Dear Reviewer of Sensors
Enclosed is the revised manuscript, entitled “A Study on Organic Semiconductor-Based Indirect X-ray Detector with Cd-free QDs for Sensitivity Improvement”. We appreciate the in-depth reviewer's comments on this manuscript, which were very helpful for improving the scientific quality of our paper. Based on the reviewer’s comments, some of the sentences were corrected in the revised manuscript. Please see the attachment.
Best Regards
Prof. Jungwon Kang

Reviewer 2 Report
Authors properly addressed most of my suggestions. The only item, where my doubt remained, is the way of the evaluation of the mobility through the SCLC model according eq. (4). The evaluation of the mobility is based on the trap-free SCLC model according ref. [47] predicting slope 2 after the fast increase of the current. Since authors derive their results from the terminal part of the curve, the slope 2 may be hardly confirmed. The validity of the model is thus unsubstantiated. Simultaneously, I am surprised from the large current density in this region, which corresponds to the power about 50mW/cm^2. Significant heating of the structure could debase the model. Also the interval of biasing, where the Trap-limited SCLC is expected, is very narrow and one may hardly believe that the SCLCs affect the shape of J-V characteristics according author's suggestions. It seems to me that based on these objections reported mobility values can be hardly accepted. Authors should consider the removal of their mobility analysis from the manuscript. I also noted that the previous value of the mobility remained in the Abstract and Conclusions.
Author Response

(The authors gave the same response as above.)
